# Chronic Periodontal Infection and Not Iatrogenic Interference Is the Trigger of Medication-Related Osteonecrosis of the Jaw: Insights from a Large Animal Study (PerioBRONJ Pig Model)

**DOI:** 10.3390/medicina59051000

**Published:** 2023-05-22

**Authors:** Matthias Troeltzsch, Stephan Zeiter, Daniel Arens, Dirk Nehrbass, Florian A. Probst, Paris Liokatis, Michael Ehrenfeld, Sven Otto

**Affiliations:** 1Department of Oral and Maxillofacial Surgery and Facial Plastic Surgery, University Hospital, LMU Munich, Lindwurmstraße 2a, 80337 Munich, Germanyparis.liokatis@med.uni-muenchen.de (P.L.);; 2Center for Oral, Maxillofacial and Facial Reconstructive Surgery, Maximilianstraße 5, 91522 Ansbach, Germany; 3AO Research Institute Davos, Clavadelerstrasse 8, 7270 Davos, Switzerland; 4MKG Probst, Sendlingerstraße 31, 80331 Munich, Germany

**Keywords:** periodontitis, MRONJ, minipig animal model, chronic inflammation, osteonecrosis, periodontitis induction, BRONJ, BRONJ Pig, PerioBRONJ Pig

## Abstract

*Background and Objectives*: Antiresorptive drugs are widely used in osteology and oncology. An important adverse effect of these drugs is medication-induced osteonecrosis of the jaw (MRONJ). There is scientific uncertainty about the underlying pathomechanism of MRONJ. A promising theory suspects infectious stimuli and local acidification with adverse effects on osteoclastic activity as crucial steps of MRONJ etiology. Clinical evidence showing a direct association between MRONJ and oral infections, such as periodontitis, without preceding surgical interventions is limited. Large animal models investigating the relationship between periodontitis and MRONJ have not been implemented. It is unclear whether the presence of infectious processes without surgical manipulation can trigger MRONJ. The following research question was formulated: is there a link between chronic oral infectious processes (periodontitis) and the occurrence of MRONJ in the absence of oral surgical procedures? *Materials and Methods:* A minipig large animal model for bisphosphonate-related ONJ (BRONJ) using 16 Göttingen minipigs divided into 2 groups (intervention/control) was designed and implemented. The intervention group included animals receiving i.v. bisphosphonates (zoledronate, n = 8, 0.05 mg/kg/week: ZOL group). The control group received no antiresorptive drug (n = 8: NON-ZOL group). Periodontitis lesions were induced by established procedures after 3 months of pretreatment (for the maxilla: the creation of an artificial gingival crevice and placement of a periodontal silk suture; for the mandible: the placement of a periodontal silk suture only). The outcomes were evaluated clinically and radiologically for 3 months postoperatively. After euthanasia a detailed histological evaluation was performed. *Results:* Periodontitis lesions could be induced successfully in all animals (both ZOL and NON-ZOL animals). MRONJ lesions of various stages developed around all periodontitis induction sites in the ZOL animals. The presence of MRONJ and periodontitis was proven clinically, radiologically and histologically. *Conclusions:* The results of this study provide further evidence that the infectious processes without prior dentoalveolar surgical interventions can trigger MRONJ. Therefore, iatrogenic disruption of the oral mucosa cannot be the decisive step in the pathogenesis of MRONJ.

## 1. Introduction

Osteonecrosis of the jaw can be induced by various medications and remains a problem for clinicians working in the field of osteology, oncology, dentistry and oral and maxillofacial surgery [1,2,3,4]. Initially, the only drugs associated with medication-related osteonecrosis of the jaw (MRONJ) were bisphosphonates (in both oral and intravenous routes of administration). However, further developments in the field of osteotropic and anti-neoplastic pharmacotherapy led to the release of an array of new drugs (from the group of targeted therapy) that can either trigger MRONJ alone or in combination with other drugs [5]. Beside bisphosphonates, the antibody denosumab, of which can be used for the same indications as bisphosphonates, is known to cause MRONJ at the same or even higher rate than bisphosphonates [6,7,8]. Furthermore, several anti-angiogenic drugs have been identified as co-factors in the evolution of MRONJ [5]. The clinical appearance of MRONJ can vary depending on its clinical stage, from pain sensations within the jaw without clinical signs, to probable bone without signs of infection, to severe stages with large areas of exposed bone, alveolar nerve or maxillary sinus involvement [4]. Both conservative and invasive treatment approaches have been suggested [9,10].

However, successful treatment of the disease can only be achieved if the pathophysiological mechanism of MRONJ is completely understood. Animal models gave initial hints that a sustained infectious process originating from a periodontal or periapical bacterial intrusion is the first and vital step for the development of MRONJ [11,12]. The inflammatory process can expand rapidly and unhampered, as antiresorptive drugs (especially bisphosphonates and denosumab) are known not only to reduce bone turnover rates, but also to alter immunologic responses, angiogenesis and epithelial cell viability [13]. These effects result in osteonecrosis and surrounding soft tissue inflammation.

Periodontitis is a widespread chronic inflammatory process that affects over 50% of adults worldwide [14]. Gingivitis and periodontitis are inflammatory diseases of bacterial origin that cause oral mucosal inflammation and periodontal bone loss [15,16]. Spontaneous MRONJ development in patients who suffered from severe forms of periodontitis have been reported anecdotally [17]. Furthermore, clinical data have revealed that periodontitis is highly prevalent in MRONJ patients [18] and that the presence and severity of periodontitis may predict the risk of MRONJ development [19,20].

The initial experiments suspected that it is mainly dentoalveolar surgical interventions (e.g., tooth extractions without soft tissue plastic closure of the wound) that trigger MRONJ [21,22,23]. However, if the above-mentioned pathogenesis theory [11] was true, then chronic infectious processes (such as periodontitis) alone without prior surgical interventions would be able to cause MRONJ. 

Very few preclinical studies in the small rodent models have investigated the association between artificially induced periodontitis and MRONJ development [24,25,26]. It was found that MRONJ could develop and exacerbate in periodontitis lesions [24,25,26]. However, the rat model bears several drawbacks, e.g., differences in bone growth patterns and histological structure compared with the human model [27,28]. These pitfalls may limit the translation of rat study results to clinical practice. 

The advantage of the Göttingen Minipig animal model is its aptitude to serve as an established periodontitis and MRONJ model [29]. Proof of principle studies for the generation of both periodontitis and MRONJ in Göttingen minipigs are available and the successful translation of animal study results to the human model has been achieved [30].

The aim of the present study was to explore the associations between periodontitis and MRONJ development in a widely accepted, preclinical animal model, the validity of which has been confirmed in human studies. Therefore, the present animal model (PerioBRONJ Pig Model) was designed. The following research question was formulated:

Can chronic infectious processes alone without prior invasive (mucosa-disrupting) measures cause medication-related osteonecrosis of the jaw (MRONJ) in an established animal model?

## 2. Materials and Methods

To address the research question a prospective animal model using Göttingen minipigs was designed. Internal review board approval was obtained and all ethical requirements regarding the conduct of large animal models were met. The conduct of the study was approved by the competent authorities of the Canton Grisons (Switzerland, application number 22/2017).

### 2.1. Animal Selection and Experimental Group Design

The study cohort consisted of 16 Göttingen minipigs (origin: Ellegaard, Denmark) that were identified by ear tags (including chips). The following inclusion criteria for the selection of the study animals were defined: (1) age, 17 months; (2) weight, 25–50 kg; (3) gender, female; (4) healthy based on clinical examination and blood analyses (examinations were carried out by board-certified veterinarians); and (5) absence of radiological changes in the jaws based on CT scans. The animals were randomly assigned to 2 groups of 8 animals each. The animals were numbered from 1 to 16 and assigned to Group 1 and Group 2 totally randomly and without a specific technique.

Group 1 was defined as the intervention group (intravenous application of antiresorptive drugs: ZOL). Group 2 animals acted as negative control group without antiresorptive treatment (NON-ZOL; Table 1). 

After the arrival of the minipigs, the pigs were given time to acclimatize (four weeks quarantine). At the same time, daily handling started with pieces of apple. Normal feeding was carried out with dry food (food: KLIBA NAFAG, Alleinfuttermittel for Minipig, 3000/PS S15). All minipigs were group-housed.

In consideration of previously published results involving the induction for an MRONJ model [12,21], the total duration of the experiment was 24 weeks.

### 2.2. Timeline of the PeriBRONJ Pig Experiment

The animal study consisted of three distinct stages: (I) antiresorptive drug pretreatment (12 weeks), (II) periodontitis induction phase (6 weeks) and (III) periodontitis progression phase (6 weeks). The antiresorptive drug pretreatment intervention was only carried out in Group 1 (ZOL). The success of the periodontitis induction intervention was confirmed after the periodontitis induction phase. The duration of the different phases was determined in concordance with previously published data from minipig models [12,29,31]. All animals were euthanized after 24 weeks. A concise overview of the experiment is provided in Table 2.

### 2.3. MRONJ Induction Protocol

**Group 1 (ZOL)** received 0.05 mg/KG Zoledronate intravenously once weekly for the whole duration of the study (24 weeks). The efficacy of zoledronate administration in minipig large animal models for MRONJ induction has been validated in previous studies [21,23,32]. Therefore, the same protocol was used. The classical MRONJ induction protocol as established in previous experiments [12,21] was altered and tooth extractions were not performed. Instead, the periodontitis induction protocol (to be explained in further detail) was carried out after 12 weeks of antiresorptive drug pretreatment. 

One week prior to euthanasia, fluorescent labeling of the bone with Oxytetracycline was performed for fluorescent quantification of the extent of necrosis as previously described.

**Group 2 (Non-ZOL)** acted as the negative control group without antiresorptive treatment. Intraoral measures to induce periodontitis in the animals were applied at the same time and in the same manner as in Group 1.

### 2.4. Definition of Gingivitis/Periodontitis

According to widely accepted criteria, gingivitis was defined as the state of acute and/or chronic inflammation of the gingiva with clinical (redness, swelling, bleeding on probing, increased sulcus fluid flow rate) and histological signs (dilated capillaries, infiltration of the gingiva with lymphocytes and plasma cells) [33,34]. Gingivitis can be considered as a preliminary phase of periodontitis [35]. 

Periodontitis is defined as an inflammatory process affecting the tooth-bearing apparatus (periodontium: tooth-retaining fibers and alveolar bone) and is associated with periodontal bone loss (attachment loss) [36]. The clinical signs of periodontitis resemble those of gingivitis plus the presence of infrabony defects.

Increased probing depth in gingivitis is not necessarily a sign of periodontitis as the probable pockets can be superior to the alveolar bone (due to gingival swelling) [37]. Radiological and histological assessments are necessary for the unequivocal distinction between gingivitis and periodontitis [38].

### 2.5. Periodontitis Induction Protocol

Prior to the periodontitis induction interventions, the periodontal status of the minipigs was documented as required by the guidelines of the American Academy of Periodontology (AAP) [15,16,37]. At baseline (12 weeks after the start of the experiment/after the antiresorptive drug pretreatment in Group 1) the following periodontal parameters were recorded at 6 randomly chosen unilateral periodontal sites and furcation involvements: periodontal probing depth (PPD), sulcus bleeding index (SBI) and bleeding on probing (BOP). Furthermore, CT scans were performed at baseline prior to the periodontitis induction intervention. 

After a comprehensive overview of the periodontal status, the next step was to induce periodontitis in the minipigs. Prior studies have revealed that the minipig model is appropriate for periodontitis induction [29]. There are two ways to induce periodontitis. Firstly, a periodontal defect can be created iatrogenically after periodontal flap elevation. To sustain gingivitis and periodontitis, a periodontal silk thread must be placed at the cementoenamel junction to increase anchorage for plaque and periodontal bacteria adhesion. The aim of this model is to supervise the progress of the already simulated periodontal defect exacerbation or improvement. This technique was used in the maxilla of all experimental minipigs (both groups: M1, Figure 1). 

The second technique to induce periodontitis is less invasive. It involves the placement of a periodontal silk suture at the cementoenamel junction (CEJ) either without or with only minimal flap elevation. While the first technique guarantees a faster and more severe periodontitis development, the second technique is more sensitive and imitates the natural course of the disease. The second model does not guarantee periodontitis induction. This technique was applied in the first mandibular molar of all animals in both groups (M1, Figure 2). Grooves at the cementoenamel junction were placed with a bur and served as a reference point for the clinical and periodontal measurements of the attachment loss (later labelled as the CEJ groove).

The periodontal defects at baseline were documented with CT scans so that the progress could be evaluated not only clinically, but also radiologically (Figure 3).

The clinical periodontal examinations, CT scans and periodontitis intervention procedures were carried out under general anesthesia (12 weeks). The following protocol was applied: premedication was injected intramuscularly (i.m.) and consisted of Ketamine (15–20 mg/kg body weight (BW)), Midazolam (0.5 mg/kg BW) and Azaperone (2 mg/kg BW). Propofol (3–5 mg/kg BW) was applied intravenously (i.v.) for anesthesia induction. The minipigs were intubated and anesthesia maintenance was achieved with inhalative Sevoflurane (ca. 1.5–2% in Oxygen) with a flow rate between 0.6 L/min and 1 L/min. The analgesic protocol comprised Carprofen for cattle (1.4 mg/kg BW i.v.), Fentanyl i.v. as needed (5–20 µg/kg) and local anesthesia with Lidocaine 2% and Bupivacaine 0.5%. For postoperative pain control, 1 shot of Buprenorphine (0.01 mg/kg) i.m. was administered immediately after surgery and Carprofen (1.4 mg/kg, i.m.) was given for 5 days postoperatively.

### 2.6. Reevaluation of Periodontitis Development and MRONJ Occurrence

The literature states that artificial/iatrogenic periodontitis induction in animal models requires at least 6 weeks of follow-up after defect creation and suture placement [29,31,39]. The dates of the follow-up examinations (clinical, periodontal and radiological) were determined accordingly and performed under general anesthesia in analogy to the above-mentioned protocol. The reevaluation exams included the detailed assessment of the maxillary and mandibular M1 regions (periodontitis induction regions) and a general examination of the remaining periodontal apparatus. 

The stage assessment of MRONJ lesions involved clinical examinations, periodontal parameters and CT scan evaluation, and was performed with respect to the valid classification system suggested by the American Academy of Oral and Maxillofacial Surgeons (AAOMS) [40].

### 2.7. Radiological Protocol

Computed tomography scans were carried out 3 times during the in vivo phase (before start of the study, after periodontitis induction and 6 weeks after periodontitis induction) of the experiment. The last CT scan was obtained after euthanasia of the animals. Bone changes around the extraction sites (region of interest) were evaluated by means of CT scans (SOMATOM Emotion 6, Siemens, Munich, Germany). The region of interest (ROI) was in line with the operated area and determined by a scout scan (Figure 4). The ROI was scanned at a nominal resolution of 0.63 mm/voxel with 0.5 mm distance between slices (120 kV 80 mA). 3D images were reconstructed using an H70s reconstruction kernel. 

### 2.8. Further Periodontitis/MRONJ Development Observation Phase and Euthanasia

After the intermediate reevaluation exam (6 weeks after the periodontitis induction intervention), the further development of periodontitis lesions and MRONJ lesions was supervised for another 6 weeks. The experiment was concluded 12 full weeks after the periodontitis induction intervention and 24 full weeks after the experiment inception. One week prior to euthanasia oxytetracycline (OT, Engemycin 10%, 25 mg/kg i.m.) was administered to achieve fluorescent bone labelling [12,40]. The pre-euthanasia examinations included clinical assessments and radiological exams (CT). Euthanasia was performed with an intravenous overdose of pentobarbiturate (300 mg/mL).

### 2.9. Post Mortem Analyses and Histological Protocol

The animals were dissected and macroscopic signs of MRONJ were documented. The mandible, maxilla and right tibia were harvested. Samples were fixed in 4% formalin, dehydrated in a graded series of ethanol and finally embedded in methylmethacrylate. The polymerized samples were sectioned using either a Leica 1600 or Exakt 310 CP diamond blade saw microtome. At least two sagittal serial sections were made, including the sutured tooth (usually maxillary and mandibular M1) and at least half of the adjacent tooth on both sides (usually the fourth premolar and the second molar) of which contact radiographs were taken. Two sections per sample were glued onto opaque plexiglass slides, ground and fine polished. Of the two sections prepared, one section was stained with Giemsa-Eosin (GE), and the other kept unstained for fluorochrome analysis. The final thickness of sections ranged from 105 to 155 µm for the Giemsa-Eosin-stained sections and 80 to 120 µm for the fluorescent label observation slides. Overview microphotographs and details of selected slides (Figure 5) were taken using a brightfield light microscope equipped with a digital camera and an image acquisition software.

The Giemsa-Eosin-stained sections were used for semi-quantitative histopathological assessment focusing on the development of MRONJ. Special focus was set on the detection of gingivitis, periodontitis, osteomyelitis, abscess formation, empyema, osteonecrosis, osteolysis and bacterial colonization. Histological findings are described, wherever possible, according to distribution (focal, multifocal and diffuse), morphologic character, and if applicable and technically possible, to severity using a semi-quantitative grading scheme (grade 0–5): grade 0 = absence of change, grade 1 = minimal/very few/very small, grade 2 = slight/few/small, grade 3 = moderate/moderate number/moderate size, grade 4 = marked/many/large and grade 5 = massive/very large number/very large size.

For the analysis of the fluorochrome-labelled bone, unstained slides were used to detect oxytetracycline-specific green fluorescence (ExMax 390 nm, EmMax 546 nm) by epifluorescence microscopy and an excitation at 405 nm by a LED light source (cooled pE-4000) and a custom-made OT filter set (Ex 391/16 nm, FT 419 nm and Em 520/16 nm). Analysis was performed semi-quantitatively using the same grading scheme as mentioned above (grade 0–5). These slides will be referred to as OT-labelled slides in the remainder of this manuscript.

The histological diagnosis of MRONJ was made when signs of orally denuded bone, osteomyelitis and/or osteonecrosis were present. Osteonecrosis was defined by foci of (>5) empty osteocytic lacunae.

### 2.10. Study Variables

The presence of periodontitis at the artificial lesions, according to the AAP criteria [15,16,37,41,42], served as the predictor variable. The clinical development of MRONJ lesions according to the criteria of the American Academy of Oral and Maxillofacial Surgeons (AAOMS) was defined as the primary outcome variable. For primary data assessment, this variable was interpreted dichotomously (no signs of MRONJ/presence of MRONJ). The exact severity of the MRONJ lesion was noted and determined as the secondary outcome variable. Further secondary outcome variables were the stage of periodontitis, the presence and severity of radiological MRONJ signs, and the presence and severity of histological MRONJ signs. All variables were coded on appropriate scales: periodontitis stage (nominal), BOP (nominal), PPD (interval), furcation involvement (nominal) and the presence, signs and severity of MRONJ (dichotomous and nominal).

### 2.11. Statistical Analysis

The gathered data were tabulated, and descriptive statistics were computed. The limited number of minipigs excluded inferential statistics. 

## 3. Results

### 3.1. Success of the Periodontitis Induction Protocol

Shallow intrabony defects could be detected at baseline in two animals of the ZOL group (Group 1, Figure 6A,B; Figure 7A,B). Intrabony defects were not found in NON-ZOL animals at baseline.

Gingivitis was induced successfully in all animals (ZOL and NON- ZOL) at both sites (maxilla and mandible). Thus, periodontitis developed regardless of the presence of an iatrogenically created bone defect in all but one mandibular site (NON-ZOL animal: Group 2, Figure 1C,D). 

The presence of gingivitis and periodontitis was checked both clinically (BOP, PPD, SBI and visible plaque accumulation: see Figure 1D) and radiographically (Figure 9 and Figure 10).

The severity of the gingivitis/periodontitis varied between the groups, with a higher burden of periodontal disease in the ZOL-group (Group 1).

The periodontal examinations revealed the following changes of PPD, BOP and SBI from baseline to the first follow-up examination (6 weeks later, average values) in ZOL and NON-ZOL animals (Table 3):

The registered increases of PPD between the baseline examinations and the follow-up examinations correlated with the radiologically recorded attachment losses. Attachment loss in the maxillary sites (iatrogenic periodontal defect) was more pronounced than in mandibular sites. The attachment loss was greater in the ZOL group (Group 1).

The periodontitis severity did not progress further during the remainder of the experiment duration.

### 3.2. Osteonecrosis Induction

#### 3.2.1. Clinical Evaluation

At the end of the experiment (24 weeks after inception, 12 weeks after the periodontitis induction intervention) clinical signs of MRONJ of different stages were visible (100%) in the ZOL group animals (Group 1). The MRONJ lesions in the maxilla (iatrogenically induced vertical periodontal defect) were of higher stages than in the mandible (only visible at the placement of silk thread within the periodontal sulcus: Table 4). 

At the first follow-up exam (6 weeks after periodontitis induction intervention, 18 weeks after inception), MRONJ lesions had developed around all maxillary sites (iatrogentic periodontal defect, Figure 8) and mandibular MRONJ could be detected in 7/8 sites (Table 4, Figure 12).

At the second follow-up examination, MRONJ was diagnosed clinically in all periodontitis induction sites of all ZOL group animals (Table 5). MRONJ developed spontaneously (without any prior iatrogenic inflammatory stimulus: Figure 9A,B) on the contralateral side in one animal. In none of the NON ZOL animals could signs of periodontal osteonecrosis be observed.

MRONJ stage migration toward higher stages was detected both in maxillary and mandibular sites between the intermediate exam and the pre-euthanasia exam.

#### 3.2.2. Histological Evaluation

Histological signs of gingivitis were present in all animals (16/16), regardless of the increased severity in the ZOL group (Group 1, Figure 10). All but one gingivitis sites had histological signs of periodontitis (Figure 11).

Periodontitis lesions in ZOL animals were more extensive. The intraoperatively placed silk thread was still in place in most sites (23/32) after euthanasia, which could serve as a proof that the thread was a causative trigger factor of periodontitis. Another important trigger factor of periodontitis was interdental food entrapment, which could be found in 27/32 sites. 

Gingival ulceration appeared in all periodontitis sites of the ZOL group. Gingival surface disruption in the NON-ZOL animals was generally mild and only histologically discernible in 50% of the sites. The destructed mucosal surface, together with the recession of the gingiva, led to orally denuded bone (mandibula: n = 7, maxilla: n = 8). This finding was only found in Group 1 sites affected with MRONJ, leading to an invasion of oral bacteria as well as the development of osteomyelitis and osteonecrosis. 

Inflammatory infiltration of the bone and bone marrow was confirmed histologically in all animals of the ZOL group (13/16 sites) and was not detected in NON-ZOL animals. These signs were markedly increased in mandibular sites. In 6/8 animals of the ZOL group, osteonecrosis was present (9/16 periodontitis sites). Viable bacterial colonies were discerned within the osteomyelitic and osteonecrotic (Figure 12 and Figure 13) areas. Osteonecrosis did not occur in NON-ZOL animals. 

The inflammatory changes in ZOL group animals were severe and disseminated, with proof of mandibular canal empyema in one animal, subdental abscesses in three animals and pulpal affection of adjacent teeth in four animals.

As expected, reduced osteoblastic activity was noted in all ZOL group animals regardless of the severity of inflammatory infiltration in both affected and pristine bone areas. In the proximity of the ARONJ areas, an increased compensatory proliferation of trabecular bone was present (Figure 14).

The analysis of the OT-labelled slides confirmed the findings gained from the HE-slides, with strongly weakened OT-labelling in osteomyelitis and osteonecrosis sites (Figure 15).

#### 3.2.3. Radiological Evaluation

The CT scans obtained immediately after the periodontitis induction intervention proved the presence of the orientation grove at the CEJ and served as a reference for the analysis of the expected attachment loss. The artificial intrabony defect in maxillary sites was 4.3 mm (+/−1.2) at baseline. In mandibular sites, the periodontal bone was level with the CEJ groove (Figure 10A). CT scans were obtained again after 6 weeks, simultaneously to the clinical reexamination. The average attachment loss was 2.7mm (+/−0.8) in the maxilla and 3.4mm (+/−1.2) in the mandible (Figure 16). 

Radiological proof of periodontal attachment loss is a defining criterion of successful periodontitis induction. Typical signs of advanced MRONJ, such as maxillary sinusitis, bone sclerosis, reactive periosteal callus apposition and sequestration could be detected in various ZOL group animals (Figure 17).

## 4. Discussion

Periodontitis as a widespread intraoral chronic inflammatory disease was considered to be the ideal inflammatory stimulus, as MRONJ is mainly an intraoral phenomenon. To address the research question, a suitable animal model for both MRONJ and periodontitis induction had to be found and both experimental arms had to be combined in a sequential manner. Prior to the experiment it was unknown whether osteonecrosis lesions could be induced by periodontitis in the minipig model, as all prior models involved traumatic tooth extractions without plastic closure of the alveolar defects [21]. A further challenge in the experimental design was the selection of the periodontitis induction model, of which different algorithms with varying invasiveness have been suggested [29,43]. The conservative placement of a periodontal ligature (the non-invasive method) for enhanced anchorage of food debris and bacteria as well as iatrogenic defect creation (involving flap elevation: the invasive method) have been proposed [44]. As this was the first experiment exploring the interrelationships of periodontitis and MRONJ, both methods were applied.

The performance of oral surgical procedures Is considered to be the main risk factor for the development of MRONJ in patient cohorts receiving antiresorptive drugs [4]. Further developments in MRONJ research have suspected that chronic inflammatory processes cause bone decay, ultimately leading to clinically visible pathology, which necessitates oral surgical procedures (e.g., tooth extractions) [10,19,21,23]. These interventions finally disrupt the oral mucosal integrity and uncover the already ongoing MRONJ pathology [45]. Recent studies have confirmed that the risk of MRONJ development is increased in patients suffering from periodontitis [46,47] and may even occur spontaneously [4]. 

The exact reasons for these “spontaneous” MRONJ cases have attracted scientific attention in the recent years. As shown in the rat model, the presence of periapical pathology in conjunction with oral surgical procedures increases the risk of MRONJ occurrence [48]. Furthermore, iatrogenically induced periodontal disease was shown to spark and exacerbate MRONJ in the rat and mouse model [24,25,49]. In turn, the prevention and treatment of periodontitis was associated with a reduction of the odds for MRONJ [25]. In summary, a significant fraction of the research exploring and confirming the significance of periodontitis on MRONJ evolution was performed in small rodent models [26]. 

For various reasons, findings from the MRONJ rat model are not entirely reliable due to marked differences between the bone homeostasis and metabolism of rats and humans. The pig model was found to cater the most solid results in preclinical MRONJ experiments [28]. Thorough literature searches failed to reveal studies investigating the influence of chronic oral inflammatory stimuli (i.e., periodontitis) on MRONJ occurrence in large mammals [26]. The present study is the first to pinpoint periodontitis (without subsequent oral surgery) as a major risk factor for MRONJ in a large animal model. These results can be transferred to clinical medicine due to the similarities between the porcine and human bone metabolism. 

In all minipigs, periodontitis was successfully induced both by the invasive method and the non-invasive method. The proof of the periodontitis induction was provided by both clinical, histological and radiological results. Radiological signs of periodontitis-induced periodontal bone loss could be observed throughout the duration of the experiment. ZOL group animals displayed a worse periodontal condition than the NON-ZOL group at baseline (before the periodontitis induction intervention), implying that ZOL infusion altered the oral health and immune status of the affected animals.

In all ZOL animals, MRONJ lesions of different severity stages were induced. In none of the NON-ZOL animals could exposed bone be diagnosed at the follow-up examinations, despite increased probing depths after successful periodontitis induction.

The MRONJ lesions in the maxilla (iatrogenically induced vertical periodontal defects) were of higher stages than in the mandible (only visible at the placement of silk thread within the periodontal sulcus). Thus, the results of the present study corroborate the hypothesis that chronic inflammatory stimuli can trigger MRONJ without iatrogenic encroachment of the mucosal integrity. This indicates that chronic infection and not oral surgical procedures is the underlying reason for MRONJ development. In none of the NON-ZOL animals could signs of periodontal osteonecrosis be observed. It may be inferred that the elimination of such infectious processes may inhibit the progression of MRONJ. In the clinical setting, MRONJ therapy involves surgical methods which necessitate mucosal integrity disruption and osteotomy: classical risk factors for MRONJ development. However, the success of MRONJ resection for complete healing of the disease has been proven in clinical studies [10].

The literature states that periodontitis develops faster in models where silk threads are placed into artificially created periodontal defects [29,50]. Therefore, infectious processes of higher severity developed in the maxilla, which sparked MRONJ of higher stages. Despite the reduced invasiveness in the mandible, both periodontitis and MRONJ evolved in those sites as well. This fact provides further proof that infectious stimuli—even without invasive procedures—can induce MRONJ in prone subjects (ZOL animals). 

Limitations of the present study must be discussed. Large animal studies require enormous infrastructural resources. Therefore, the number of experimental animals was limited to 16 (8 animals in each group). While this number may seem low, it is comparable and even greater than the total number of animals included in most other large vertebrate MRONJ studies where usually a maximum of 12 animals are included [26]. However, some MRONJ studies in small vertebrate models involve over 100 experimental animals [26]. 

Due to the limited amount of experimental animals in this study, the gathered data could only be analyzed in a descriptive manner. The threshold for reasonable and reliable calculation of inferential statistics should be set at 10 animals per group. Therefore, the results of the study may be considered as preliminary. 

The results of the study convey clinical significance, which could be shown by the descriptive analysis. Periodontitis could be induced in all animals of the intervention and the control group. MRONJ was induced in all experimental sites of the ZOL group animals, regardless of the periodontitis induction method used (invasive/non-invasive). In one animal, a spontaneous MRONJ lesion was diagnosed in a pristine jaw site (opposite side of the jaw) which further corroborates the hypothesis that the chronic infectious stimuli are decisive in the pathogenesis of MRONJ. Had an inferential statistical analysis with Fisher’s exact test been implemented, significant results would have been computed.

The creation of an iatrogenic periodontal crevice in the maxilla might have functioned as an artificial trauma and triggering factor for MRONJ. Critics of the experiment might compare this clinical setting with the situation after dentoalveolar surgery because the bone was processed with a bur after elevation of a mucosal flap. These allegations are invalid for various reasons: (1) the creation of a periodontal crevice cannot be considered as dentoalveolar surgery in particular, (2) in all cases the periodontal defects were covered by plastic closure of the flap (there is evidence from the BRONJ-Pig 2 experiment that MRONJ development can be prevented by plastic closure of the alveolar bone after traumatic extraction) [21], (3) MRONJ lesions developed with the same reliability in the mandibular sites in which the periodontitis induction was non-invasive and (4) even spontaneous MRONJ eruptions were observed. Furthermore, histopathological proof of infectious bone changes was greater in mandibular sites in which the non-invasive periodontitis induction protocol was applied.

## 5. Conclusions

The present (PeriBRONJ) pig experiment was designed to confirm the theory that infectious stimuli are the trigger of MRONJ. The results of the experiments showed that MRONJ can be induced by periodontitis regardless of its induction method (iatrogenic crevice creation with periodontal suture placement/periodontal suture placement only), or even spontaneously in pristine jaw areas. Therefore, this experiment provides further evidence that MRONJ is caused by chronic infectious processes and can develop without preceding dentoalveolar surgical interventions.

## Figures and Tables

**Figure 1 medicina-59-01000-f001:**
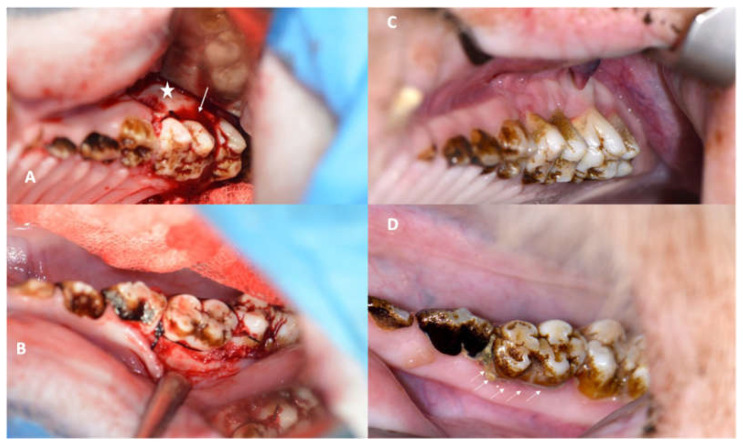
(**A**): Intraoperative view of the first maxillary molar with an iatrogenically induced anterior periodontal crevice (defect: star) and periodontal silk suture in place (arrow). (**B**): Silk suture in place around the first mandibular minipig molar achieved by only minimal flap elevation. (**C**): Image showing the first maxillary molar in a NON-ZOL pig after 6 weeks of periodontitis induction; the retraction and redness of the gingiva can be observed. (**D**): Image of a mandibular molar in a NON-ZOL minipig after 6 weeks of periodontitis induction; the excessive accumulation of plaque can be observed as a sign of pocket secretion (arrows).

**Figure 2 medicina-59-01000-f002:**
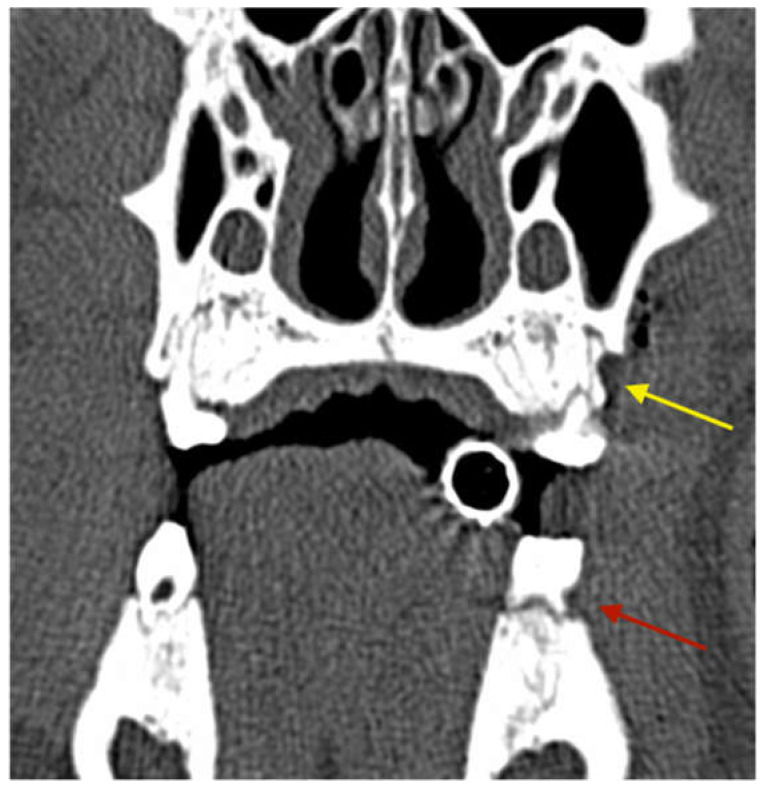
CT scan obtained directly after the surgery showing the artificial vertical periodontal defect (**yellow arrow**: maxilla) and the groove in the first mandibular molar (**red arrow**: mandible).

**Figure 3 medicina-59-01000-f003:**
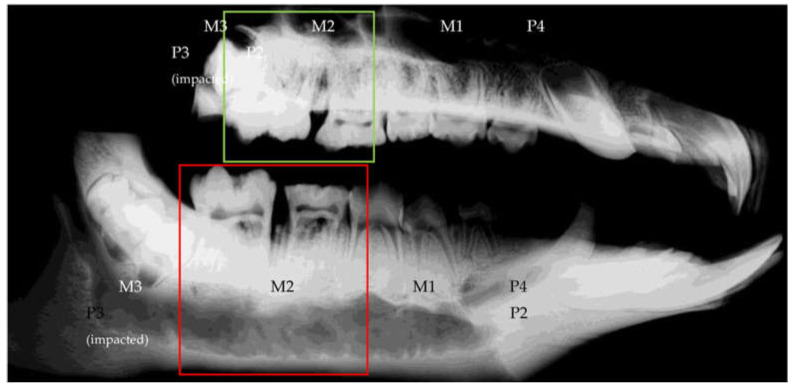
Scout scan of the ROI showing upper and lower jaw in complete overview (**top**: left hemi-maxilla; **bottom**: left hemi-mandibula).

**Figure 4 medicina-59-01000-f004:**
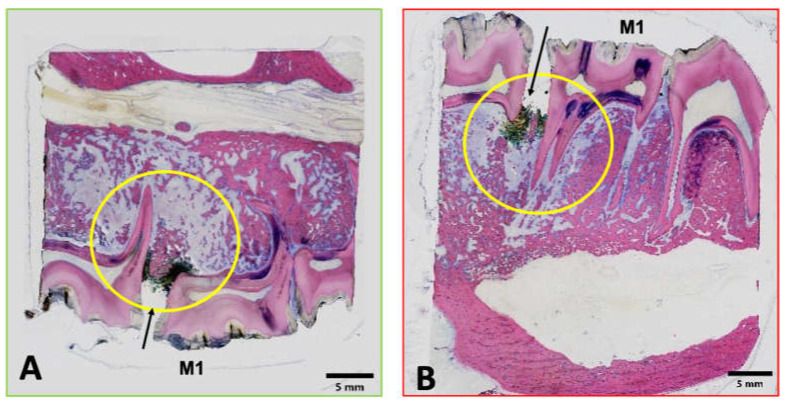
Digital microphotographs of GE-stained slides (green (**A**): maxilla; red (**B**): mandibula) showing the areas assessed by histopathological analysis. The yellow circle highlights the area of interest.

**Figure 5 medicina-59-01000-f005:**
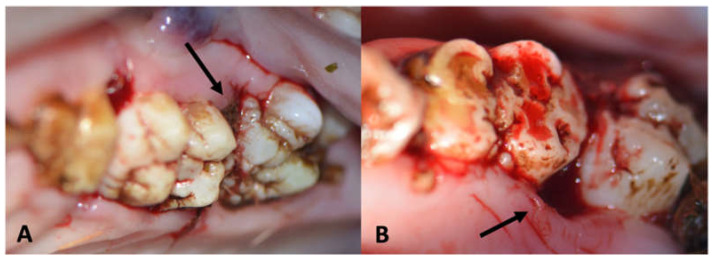
Images of ZOL group animals with periodontal intrabony defects at baseline ((**A**): maxillary fist molar; (**B**): mandibular first molar). Black arrows indicating sites of periodontal defects.

**Figure 6 medicina-59-01000-f006:**
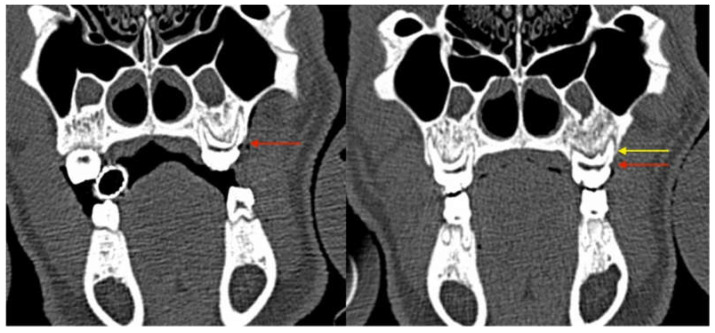
Coronal CT sections showing progressive bone loss at the periodontitis induction site (maxilla, directly postoperative: the red arrow marks the CEJ groove; the yellow arrow marks the vertical bone loss).

**Figure 7 medicina-59-01000-f007:**
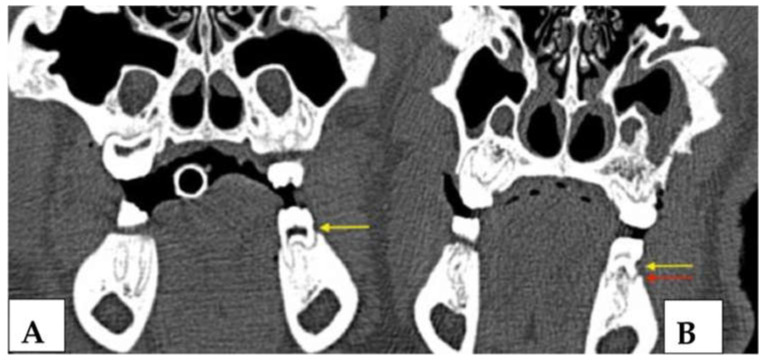
Coronal CT sections of the identical intraoral site (left mandibular first molar) immediately after CEJ groove creation and silk suture placement (left image, (**A**) yellow arrow marks the crestal bone at baseline) and at the first follow-up appointment 6 weeks later (February 2018, right image, (**B**) red arrow marks the crestal bone after periodontitis induction). The vertical bone loss can be observed which proves the successful induction of periodontitis. At the maxillary site radiologic signs of advanced-stage MRONJ (sequestrum of the facial bony sinus wall and chronic sinus mucosal swelling) are present.

**Figure 8 medicina-59-01000-f008:**
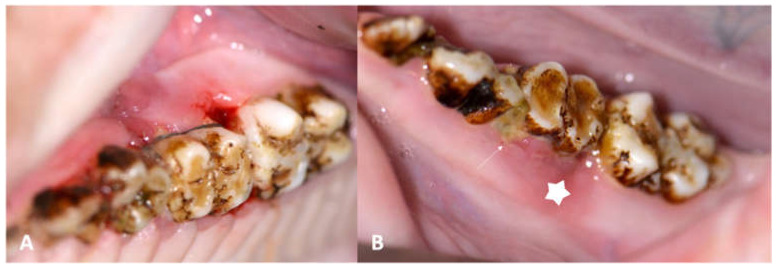
(**A**): Image of maxillary molars in a ZOL minipig showing inflammatory signs of the gingiva, excessive gingival retraction and intrabony defect formation both on the mesial aspect (iatrogenically induced) and the distal aspect (newly formed: arrows); the periodontal pockets showed pus excretion (MRONJ stage 2). (**B**): Image of mandibular molars in a ZOL minipig showing inflammatory signs of the gingiva, excessive plaque accumulation (sign of periodontitis: arrow); the bony surface could be probed on the distal aspect of the molar (MRONJ stage 1: star).

**Figure 9 medicina-59-01000-f009:**
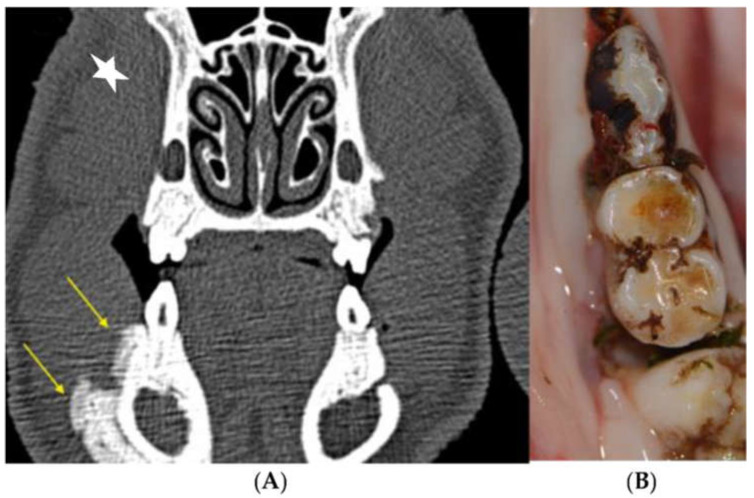
Mandibular spontaneous MRONJ development. (**A**) The lesions were primarily detected on scheduled CT scans because of the massive periosteal callus formation, as a sign of chronic bony inflammation and increased turnover. (**B**) Clinically, a massive vertical bony defect on the distal aspect of the first mandibular molar could be observed.

**Figure 10 medicina-59-01000-f010:**
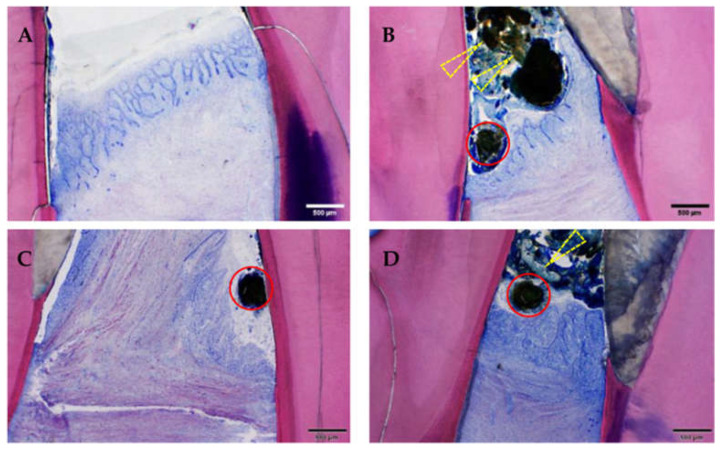
Suture material, food material, gingivitis and periodontitis. Note the presence of (mesially cut) suture material (red circles) and bacterially contaminated food material (yellow arrows), as well as the chronically inflamed gingiva of differing severity grades: (**A**) minimal, (**B**) slight, (**C**) moderate and (**D**) severe. Moreover, the gingiva is recessed below the enamel–dentin border leading to inflammation of the dental cement and of the periodontal ligament (periodontitis). In all four samples, the basal membrane is continuous (=no ulceration). GE-stained slide, magnification: (**A**–**D**) scale bar 500 μm (objective 4×).

**Figure 11 medicina-59-01000-f011:**
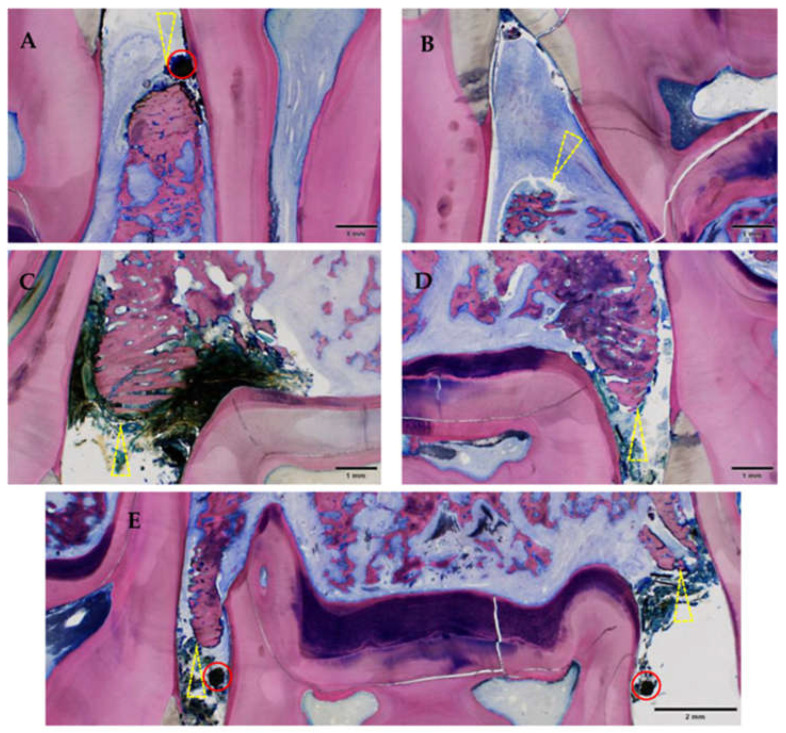
Gingival ulceration and orally denuded bone (all affected sites in this study). Note the discontinuity of the mucosa (ulceration) combined with orally denuded bone (yellow triangle) leading to an entry point for oral flora. Mandible (**A**,**B**); maxilla (**C**–**E**); all of Group 1 (ZOL-treated), GE-stained slide, magnification: (**A**–**E**) scale bar 1 mm (objective 2×)), scale bar 2 mm (objective 2×, image stitched); the silk suture is still in place (red circle).

**Figure 12 medicina-59-01000-f012:**
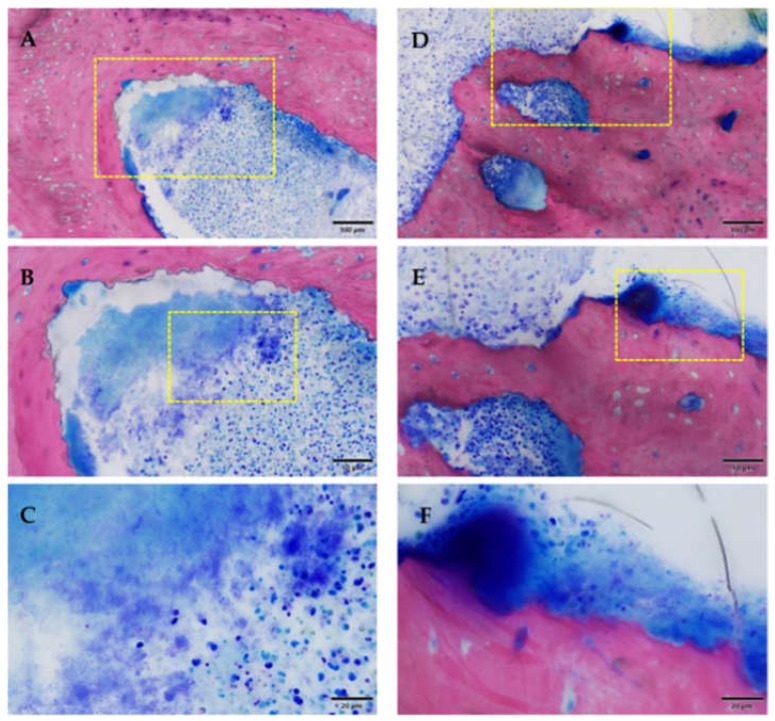
Osteomyelitis and bacterial infection, the yellow box marks the area of magnification. (**A**–**C**) area remote from ulceration; (**D**–**F**) area near ulceration. Note the presence of karyorrhectic granulocytic cells and cell debris (**A**–**C**), or additional lymphoplasmacytic cells (mixed inflammation). Inflammation with coccoid bacteria could be seen in both samples (**C**,**F**). Samples of Group 1 [ZOL-treated], GE-stained slide, magnification: (**A**,**D**) 100 μm (objective 20×), (**B**,**E**) scale bar 50 μm (objective 40×), (**C**,**F**) scale bar 20 μm (objective 100× oil).

**Figure 13 medicina-59-01000-f013:**
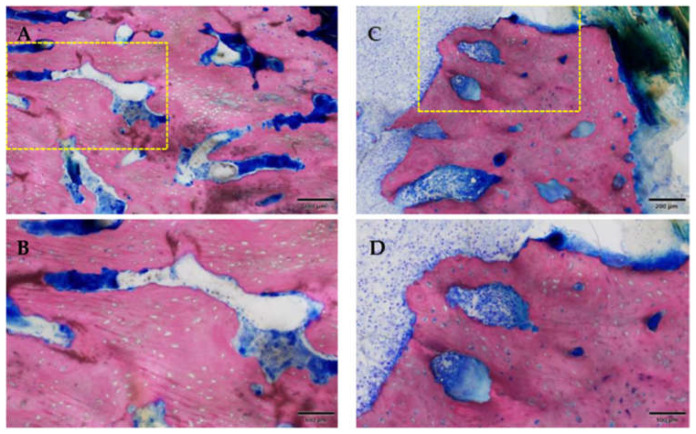
Osteonecrosis of the bone stock (superficial near the ulceration; the yellow box marks the area of magnification). Note the empty osteocytic lacunae defining dead bone tissue. Group 1 (ZOL-treated), GE-stained slide, magnification: (**A**,**C**) 200 μm (objective 10×), (**B**,**D**) scale bar 100 μm (objective 20×).

**Figure 14 medicina-59-01000-f014:**
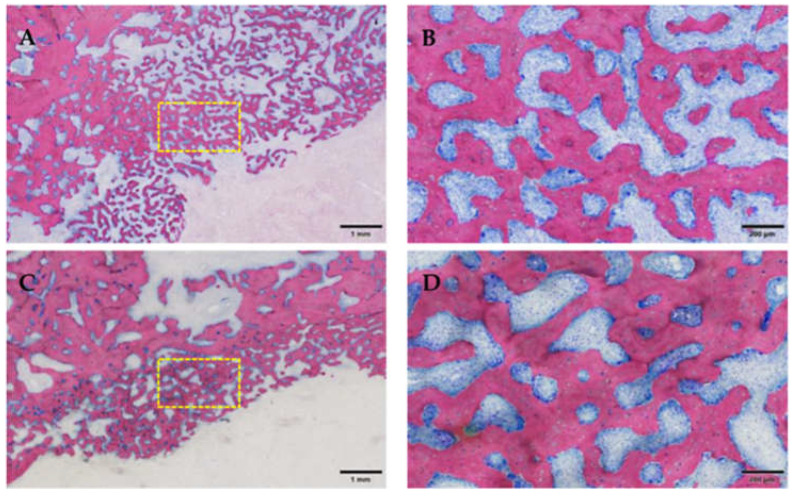
Proliferation of trabecular bone, the yellow box marks the area of magnification. (**A**,**C**): Note the new formation of bone near areas of osteomyelitis, and a low amount of bone at the inflamed and infected alveolar bone area. (**B**,**D**): In contrast to the usually more compact bone of the alveolar bone stock, this bone is of the trabecular type and exhibits high remodeling activity with both osteoblasts and osteoclasts on its surface. Group 1 (ZOL-treated), GE-stained slide, magnification: (**A**,**C**) scale bar 1 mm (objective 2×), (**B**,**D**) 200 μm (objective 10×).

**Figure 15 medicina-59-01000-f015:**
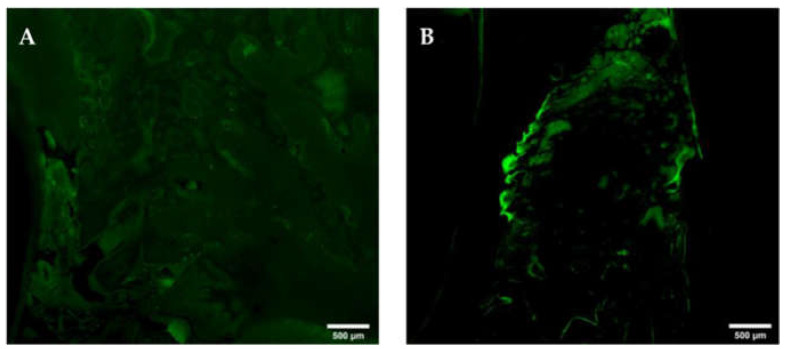
Oxytetracycline-labelling at interdental areas with orally denuded, osteonecrotic bone. Note the decrease or complete loss ((**A**), yellow circle) of OT-labelling at areas of orally denuded and osteonecrotic bone. Mandibula (**A**); maxilla (**B**) G. All of Group 1 (ZOL-treated), OT-labelled slide, epiFL illumination, magnification: A–H scale bar 500 μm (objective 10×, image stitched).

**Figure 16 medicina-59-01000-f016:**
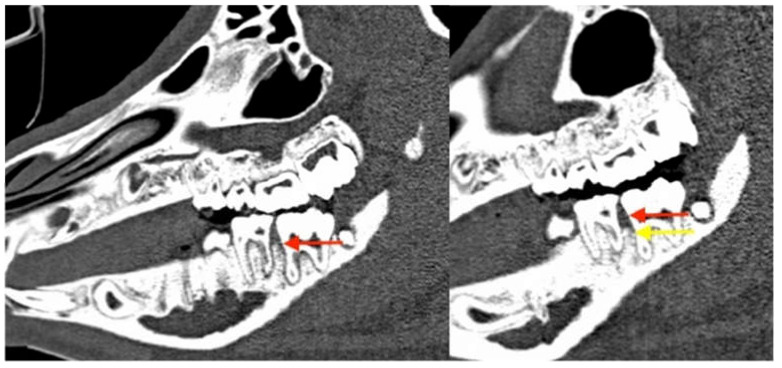
Corresponding sagittal CT scans. The red arrows mark the position of the alveolar bone after 6 weeks of follow-up, with clear clinical but no radiological signs of periodontitis and MRONJ at a mandibular site. The yellow arrow shows the dramatic vertical bone loss that occurred between week 6 and 12 of the experiment.

**Figure 17 medicina-59-01000-f017:**
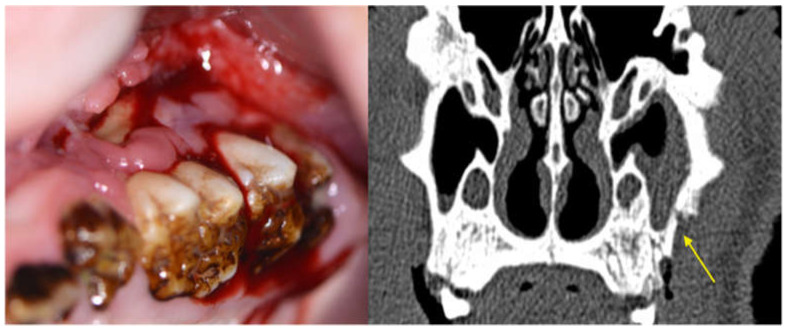
Image of maxillary molars in a ZOL minipig showing inflammatory signs of the gingiva, excessive gingival retraction and exposed bone; the symptomatology was accompanied by malodor. The CT scan shows sinus involvement and sequestrum formation (yellow arrow, MRONJ stage 3).

**Table 1 medicina-59-01000-t001:** Group allocation.

Group	Duration of Experiment	Number of Animals
Group 1: ZOL	24 weeks	8
Group 2: NON-ZOL	24 weeks	8
Total		16

**Table 2 medicina-59-01000-t002:** Experiment overview.

Time	Action	All Minipigs	
4 weeks before experiment inception (-4 weeks)	Arrival of Animals	Quarantine	
Inception of experiment (0 weeks)	Assignment to groups	Group 1 (n = 8) Zoledronate group	Group 2 (n = 8) Control group (no zoledronate)
Pretreatment phase (12 weeks)	Induction protocol of BRONJ-Pig Experiment [11]	0.05 mg/KG body weight zoledronate iv weekly	No i.v. treatment
Baseline	Periodontitis induction	Maxillary defect + suture; Mandibular suture	Maxillary defect + suture; Mandibular suture
Progression time (6 weeks)	Assessment exam	Intraoral exam, CT	Intraoral exam, CT
Progression/observation time (6 weeks)	Assessment exam	Intraoral exam, CT, euthanasia	Intraoral exam, CT, euthanasia

**Table 3 medicina-59-01000-t003:** Development of periodontal parameters over time in NON-ZOL and ZOL animals; increased values indicate periodontitis development.

	Group 1 (ZOL)	Group 2 (NON-ZOL)	Group 1 (ZOL)	Group 2 (NON-ZOL)
Parameter	Baseline (December 2017)	Baseline (December 2017)	Follow-up (February 2018)	Follow-up (February 2018)
PPD	7 mm	4 mm	13 mm	9 mm
BOP	42%	35%	100%	75%
SBI	53%	45%	100%	100%

**Table 4 medicina-59-01000-t004:** MRONJ stages after 6 weeks of periodontitis induction.

MRONJ Stage	Maxilla	Mandible
I	1	3
II	3	4
III	4	0
Total	8	7

**Table 5 medicina-59-01000-t005:** MRONJ stages after 12 weeks of periodontitis induction.

MRONJ Stage	Maxilla	Mandible
I	0	1
II	2	5
III	6	2
Total	8	8

## Data Availability

Data is contained within the article or graphical abstract.

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
