# Peer review of "Chronic Periodontal Infection and Not Iatrogenic Interference Is the Trigger of Medication-Related Osteonecrosis of the Jaw: Insights from a Large Animal Study (PerioBRONJ Pig Model)"

_medicina, 2023, doi:10.3390/medicina59051000_

Round 1

Reviewer 1 Report

The study by Troeltzsch and coworkers is a good and ideal study in its field. The introduction is good. However, the abstract and the introduction sections are so long. please improve.

Please also remove the specific aims from the introduction section.

The major point are also exist:

The lack of sample size, the lack of inclusion and exclusion criteria.

The statistical methods are non complete. It is so important in your conclusion.

Based on the new modifications, please change the conclusion section as well.

Author Response

The study by Troeltzsch and coworkers is a good and ideal study in its field. The introduction is good. However, the abstract and the introduction sections are so long. please improve.

The authors would like to thank the reviewer for the effort and time spent in reviewing and evaluating the present manuscript.

The abstract was substantially shortened. The introduction was adapted and slightly shortened as requested. The study is elaborate and complex. Therefore, the authors would like to enable the readers to understand the full scope of underlying research which was the reason for an extensive introduction.

Please also remove the specific aims from the introduction section.

As requested, the specific aims were removed.

The major point are also exist:

The lack of sample size, the lack of inclusion and exclusion criteria.

The present study is a large animal study. The problem of the limited sample size is already mentioned in the discussion section. The present study involves the largest sample size of all MRONJ large animal studies.

Inclusion and exclusion criteria do not apply in the study setting. The parameters for the selection of the study animals are stated in the M&M section.

The statistical methods are non complete. It is so important in your conclusion.

Due to the limited number of study animals inferential statistics were not computed. The strengths/weaknesses paragraph in the discussion was extended accordingly.

Based on the new modifications, please change the conclusion section as well.

The conclusion was adapted as advised.

Reviewer 2 Report

‘Can chronic inflammatory processes alone without prior invasive (mucosa disrupting) measures cause medication-related osteonecrosis of the jaw (MRONJ) in an established animal model?’

Abstract

-       Abstract should be shortened. Especially, the ‘Background and Objectives’ section is too long.

Introduction

The content of this section is excellent. However, the order of the content is a little confusing. The topic should explain the issue from the general to the specific.

-       I suggest adding some pathophysiological factors after the first paragraph. After a short explanation of pathophysiological factors, periodontitis could be elaborated as a MRONJ-causing factor.

-       After that, the need for experimental models for MRONJ pathogenesis

-       Two paragraphs (Lines 86-98) could be transported after this section. After mentioning small animal models, their usage in MRONJ pathogenesis, and the disadvantages of small animal models, Göttingen minipigs and their advantages especially on this topic could be mentioned.

Methods

-       What was the randomization technique?

-       The dosages and experiment design have been well described. This study is very detailed and contains many steps and different evaluation methods at different times. Only readable material is confusing. I suggest that a timeline visual could be created. In this way, the material method and also indirect results sections would be more clear for the readers.

-       My suggestion for visual elements (also in the Results section) is that composite figures could be figured out. In this way, the visual data could be comparable. Figure 1-2, Figure 7-8, Figure 11-12.

-       Histological method was well described

-       Overall, this is a very detailed study, the methods are very clear and the results are supporting the hypothesis. However, the biggest issue is that there is not any statistical analysis. The authors explained this as ‘the limited number of minipigs excluded inferential statistics.’ It is understandable using fewer specimens when conducting research with large animals, but lacking statistical analysis is a crucial deficiency. Because of that the findings of the study could be accepted as preliminary.

Results

-       Results are clear and well described. I suggest uniting tables, which are explaining the results of the same parameter in different timelines (e.g. Table 2-3; Table). In this way, the data would be more understandable and comparable for the readers.

Discussion

-       In my opinion, there isn't any need for the first sentence repeating the aim of the study (Lines 479-482). The second sentence is more suitable for a beginning sentence. It is more general.

-       These three paragraphs (Lines 519-534) would stand better at the beginning of the discussion section.  In these three paragraphs, you mentioned the importance of chronic inflammation in the pathophysiology of MRONJ, spontaneous MRONJ formation, and periodontitis. In the continuation, you also explained rat models and their disadvantages. In my opinion, you should explain the advantages of your large animal model after this section. And then the challenges of such a model and how you cope with these challenges.

Author Response

‘Can chronic inflammatory processes alone without prior invasive (mucosa disrupting) measures cause medication-related osteonecrosis of the jaw (MRONJ) in an established animal model?’

Abstract

-       Abstract should be shortened. Especially, the ‘Background and Objectives’ section is too long.

The authors would like to thank the reviewer for the effort and time spent in reviewing and evaluating the present manuscript.

As requested, the abstract was substantially shortened.

Introduction

The content of this section is excellent. However, the order of the content is a little confusing.

The authors thank the reviewer for carefully reviewing the introduction.

The topic should explain the issue from the general to the specific.

-       I suggest adding some pathophysiological factors after the first paragraph. After a short explanation of pathophysiological factors, periodontitis could be elaborated as a MRONJ-causing factor.

-       After that, the need for experimental models for MRONJ pathogenesis

-       Two paragraphs (Lines 86-98) could be transported after this section. After mentioning small animal models, their usage in MRONJ pathogenesis, and the disadvantages of small animal models, Göttingen minipigs and their advantages especially on this topic could be mentioned.

As requested, the introduction section was rearranged.

Methods

-       What was the randomization technique?

No specific randomization technique was used.

-       The dosages and experiment design have been well described. This study is very detailed and contains many steps and different evaluation methods at different times. Only readable material is confusing. I suggest that a timeline visual could be created. In this way, the material method and also indirect results sections would be more clear for the readers.

As requested a concise table-like experiment overview was added (table 2).

-       My suggestion for visual elements (also in the Results section) is that composite figures could be figured out. In this way, the visual data could be comparable. Figure 1-2, Figure 7-8, Figure 11-12.

The figures were rearranged as requested.

-       Histological method was well described

-       Overall, this is a very detailed study, the methods are very clear and the results are supporting the hypothesis. However, the biggest issue is that there is not any statistical analysis. The authors explained this as ‘the limited number of minipigs excluded inferential statistics.’ It is understandable using fewer specimens when conducting research with large animals, but lacking statistical analysis is a crucial deficiency. Because of that the findings of the study could be accepted as preliminary.

The authors would like to thank the reviewer for mentioning this important aspect. Adding a “true” statistical analysis would only be possible if more data were available. The study is among the most extensive animal studies of its kind in the realm of MRONJ research. The preliminary intention of the study was mentioned in the discussion section.

Results

-       Results are clear and well described. I suggest uniting tables, which are explaining the results of the same parameter in different timelines (e.g. Table 2-3; Table). In this way, the data would be more understandable and comparable for the readers.

As requested the tables were rearranged.

Discussion

-       In my opinion, there isn't any need for the first sentence repeating the aim of the study (Lines 479-482). The second sentence is more suitable for a beginning sentence. It is more general.

-       These three paragraphs (Lines 519-534) would stand better at the beginning of the discussion section.  In these three paragraphs, you mentioned the importance of chronic inflammation in the pathophysiology of MRONJ, spontaneous MRONJ formation, and periodontitis. In the continuation, you also explained rat models and their disadvantages. In my opinion, you should explain the advantages of your large animal model after this section. And then the challenges of such a model and how you cope with these challenges.

The authors would like to thank the reviewer for the valuable advice. The discussion was rearranged as recommended.

Reviewer 3 Report

This article is a further study of the pathogenicity factors of MRONJ, the details of which are not yet known. In particular, the results of this study, which fundamentally examined whether surgical invasion or inflammatory factors were responsible for the onset of MRONJ, are clinically valuable and may shed light on the treatment of MRONJ patients. Furthermore, the obtained results of this animal study are of great clinical value and will be a major foundation for future MRONJ-related research. However, there are some expressions that are difficult for readers to understand, so the following points could be improved to make the research report even better.

 Material and Methods section

Method of creating periodontal defects in the PerioBRONJ pig model.

Can we assume that the experimental group was not formed with a bur during the open flap and that it occurred spontaneously, similar to periodontitis in humans? If so, it would be clearer to use the clear statement that the bur did not form the bone defect spontaneously. If the burs were used to form bone defects, they would be "artificial iatrogenic interference" and would contradict the hypothesis of this study.

 Why did the authors choose zoledronate among the many BPs? The author states referring to previous experimental methods, but it would be desirable to provide a reason. In addition, the content will be clearer if the method of administration and volume of the drug used in the experimental model is reasonable with respect to the method of administration in humans.

 Supplementary explanations should be added on these points.

Discussion section

 Based on clinical, histological, and radiological results, the paper concludes that inflammation alone may be the pathogenic factor. The authors' predictions and considerations regarding the mechanism of inflammation on lesion progression, either pathologic or bacteriologic, should also be included in this section.

Author Response

Reviewer 3:

This article is a further study of the pathogenicity factors of MRONJ, the details of which are not yet known. In particular, the results of this study, which fundamentally examined whether surgical invasion or inflammatory factors were responsible for the onset of MRONJ, are clinically valuable and may shed light on the treatment of MRONJ patients. Furthermore, the obtained results of this animal study are of great clinical value and will be a major foundation for future MRONJ-related research. However, there are some expressions that are difficult for readers to understand, so the following points could be improved to make the research report even better.

 Material and Methods section

Method of creating periodontal defects in the PerioBRONJ pig model.

Can we assume that the experimental group was not formed with a bur during the open flap and that it occurred spontaneously, similar to periodontitis in humans? If so, it would be clearer to use the clear statement that the bur did not form the bone defect spontaneously. If the burs were used to form bone defects, they would be "artificial iatrogenic interference" and would contradict the hypothesis of this study.

The authors would like to thank the reviewer for bringing up this important aspect. Indeed, the maxillary periodontal defects were created with a bur. BUT: in the mandibular site, NO encroachment on the bony integrity was performed.

The discussion includes a paragraph elaborating on this exact problem

“The creation of an iatrogenic periodontal crevice in the maxilla might have functioned as an artificial trauma and triggering factor for MRONJ. Critics of the experiment might compare this clinical setting with the situation after dentoalveolar surgery because the bone was processed with a bur after elevation of a mucosal flap. These allegations are invalid for various reasons: (1) the creation of a periodontal crevice cannot be considered as dentoalveolar surgery in particular, (2) in all cases the periodontal defects were covered by plastic closure of the flap (there is evidence from the BRONJ-Pig 2 experiment that MRONJ development can be prevented by plastic closure of the alveolar bone after traumatic too extraction) [21], (3) MRONJ lesions developed with the same reliability in the mandibular sites in which the periodontitis induction was non-invasive and (4) even spontaneous MRONJ eruptions were observed. Furthermore, histopathological proof of inflammatory bone changes were greater in mandibular sites in which the non-invasive periodontitis induction protocol was applied.“

 Why did the authors choose zoledronate among the many BPs? The author states referring to previous experimental methods, but it would be desirable to provide a reason. In addition, the content will be clearer if the method of administration and volume of the drug used in the experimental model is reasonable with respect to the method of administration in humans.

The authors were eager to use validated methods in the basic experimental design. Therefore, the previously described method of MRONJ induction described by Pautke et al. in 2011 was used. If a divergent protocol had been adopted, the clinical efficacy should have been proven in a preceding experiment. For financial and logistic reasons validated protocols were applied. This aspect was added to the M&M section.

 Supplementary explanations should be added on these points.

Discussion section

 Based on clinical, histological, and radiological results, the paper concludes that inflammation alone may be the pathogenic factor. The authors' predictions and considerations regarding the mechanism of inflammation on lesion progression, either pathologic or bacteriologic, should also be included in this section.

The discussion section was extended to address this very important aspect.

Round 2

Reviewer 1 Report

Thanks for your modifications and responses. However, the sample size and statistical method are needed. I did not justify by your reply.

Reviewer 2 Report

Thank you for submitting the revised version of the manuscript.